# Rapid Foreign Object Detection System on Seaweed Using VNIR Hyperspectral Imaging

**DOI:** 10.3390/s21165279

**Published:** 2021-08-04

**Authors:** Dong-Hoon Kwak, Guk-Jin Son, Mi-Kyung Park, Young-Duk Kim

**Affiliations:** 1ICT Research Institute, DGIST, Daegu 42988, Korea; gns9452@dgist.ac.kr (D.-H.K.); sudopop@dgist.ac.kr (G.-J.S.); 2School of Food Science and Biotechnology, Kyungpook National University, Daegu 41566, Korea; parkmik@knu.ac.kr

**Keywords:** foreign object detection, hyperspectral imaging, visible and near-infrared, spectroscopy, signal processing, seaweed

## Abstract

The consumption of seaweed is increasing year by year worldwide. Therefore, the foreign object inspection of seaweed is becoming increasingly important. Seaweed is mixed with various materials such as laver and sargassum fusiforme. So it has various colors even in the same seaweed. In addition, the surface is uneven and greasy, causing diffuse reflections frequently. For these reasons, it is difficult to detect foreign objects in seaweed, so the accuracy of conventional foreign object detectors used in real manufacturing sites is less than 80%. Supporting real-time inspection should also be considered when inspecting foreign objects. Since seaweed requires mass production, rapid inspection is essential. However, hyperspectral imaging techniques are generally not suitable for high-speed inspection. In this study, we overcome this limitation by using dimensionality reduction and using simplified operations. For accuracy improvement, the proposed algorithm is carried out in 2 stages. Firstly, the subtraction method is used to clearly distinguish seaweed and conveyor belts, and also detect some relatively easy to detect foreign objects. Secondly, a standardization inspection is performed based on the result of the subtraction method. During this process, the proposed scheme adopts simplified and burdenless calculations such as subtraction, division, and one-by-one matching, which achieves both accuracy and low latency performance. In the experiment to evaluate the performance, 60 normal seaweeds and 60 seaweeds containing foreign objects were used, and the accuracy of the proposed algorithm is 95%. Finally, by implementing the proposed algorithm as a foreign object detection platform, it was confirmed that real-time operation in rapid inspection was possible, and the possibility of deployment in real manufacturing sites was confirmed.

## 1. Introduction

Dried seaweed is made by mixing various seaweeds (e.g., laver and sargassum fusiforme) and drying them. It is one of the representative side dishes in South Korea. Seaweed is a well-being food rich in minerals, vitamins, and proteins and low in calories. The Wall Street Journal in the United States reported that it is a Korean superfood [1]. Korean seaweed is an Asian seaweed standard adopted by the Codex Alimentarius Commission. It currently exports to more than 100 countries, with annual exports worth $600 million. This is a steadily increasing figure in recent years As it is gaining popularity around the world, hygiene management is also becoming important when manufacturing seaweed.

Since seaweed is mixed with materials and dried during the manufacturing process, there is a high possibility of foreign objects coming out. There are various kinds of foreign objects. Foreign objects such as shrimp shells are harmless even if consumed. However, foreign objects such as screws and stones can damage teeth, and bugs can be at risk of bacterial infection. To prevent this accident, it is very important to detect foreign objects through non-destructive inspection. Seaweed is not easy to detect foreign objects. Because it has various colors even in the same seaweed and has uneven and greasy surface. Due to these characteristics, according to factory officials, conventional foreign object detectors currently used in real manufacturing sites have less than 80% accuracy.

In the food industry, various kinds of foreign object detectors are used. Examples include X-ray detectors, metal detectors, infrared (IR) detectors, and RGB image-based detectors [2,3,4,5]. These detectors are slightly different from each other for application purposes. X-ray detectors and metal detectors can only detect foreign objects in certain materials (metal or hard material), and internal inspection is also possible. However, their main drawback is that they cannot detect soft foreign objects such as insects, mold, etc. Meanwhile, the RGB-based detector and IR detector can detect such soft foreign objects by analyzing the color and infrared spectral characteristics of the adsorbed material on the food surface. However, they still have poor performance against foreign objects having a similar color with that of food.

Recently, the detector using hyperspectral imaging is specialized for surface inspection with highly sophisticated spectral analysis that is superior to conventional IR detectors. Hyperspectral imaging technique acquires images in the form of the n-dimensional hypercube. It acquires data not only in the spatial direction but also in the wavelength direction. In addition, these images contain wavelength-specific spectral information, ranging from dozens to hundreds per pixel. Through such advantages, hyperspectral imaging is being actively researched in various fields of the food industry [6,7,8,9,10], such as foreign object detection and quality inspection. Most of the previous works did not explicitly present the real-time processing speed. There are few works that can inspect food on a conveyor belt moving at high speeds in real-time. Moreover, the previous works were to detect only specific foreign objects or inspect food quality in specific situations. However, in the food manufacturing industry, it is necessary to satisfy their production speed and to prepare for unspecified foreign objects. To the best of our knowledge, real-time analysis of seaweed using conveyor belts and hyperspectral imaging is the first study.

In this study, in order to resolve the above-mentioned problem of hyperspectral imaging, we firstly adopt visible and near-infrared (VNIR) hyperspectral imaging techniques [11,12,13]. The VNIR uses wavelength-specific analysis from visible light ranges (400–750 nm) to near-infrared ranges (750–1000 nm). When the camera operates at the full band (224 bands per pixel), full resolution (1024 pixels per line), and full fps (Frames Per Second) (330 fps), it needs to process 229,376 bytes of data per line and 68,812,800 bytes of data per second. However, it is still difficult to process this vast amount of data in real-time with an accurate inspection. This is the main reason for the absence of a commercialized hyperspectral image-based foreign object detection system, especially in conveyor moving inspection. In order to tackle this problem, the characteristics of the seaweed surface should be considered. In the case of data with non-linear values such as seaweed surface pixels, a deep neural network algorithm (e.g., Convolutional Neural Network (CNN), Region-based Convolutional Neural Network (R-CNN), You Only Look Once (YOLO), etc.) [14,15,16,17] can be used for classification and detection. Detection algorithms such as R-CNN and YOLO are frequently used for foreign object detection, but it is difficult to process the spectral dimension of hyperspectral images in a real-time operation. There are various methods for detecting foreign objects in hyperspectral images using CNN, including pixel-wise classification using 1D CNN, 2D CNN through dimensionality reduction, and hypercube classification using 3D CNN [18,19]. These methods allow real-time processing when conveyor belts move at low speeds, but cannot process real-time inspections at the speed required by recent commercial seaweed manufacturers. Commercial seaweed manufacturers require more rapid inspection with a conveyor belt speed of at least 30 cm/s, which can inspect one piece of seaweed per second. Thus, in order to detect a foreign object of about 1 mm width, it should be inspected in the environment at least 300 fps because when the conveyor belt moves 30 cm/s, the camera needs to scan 300 lines/s to have a horizontal resolution of 1 mm. The proposed algorithm focuses on real-time inspections of conveyor belt speed required by recent commercial seaweed manufacturers with accuracy inspection.

The rest of this paper is categorized into four chapters. Section 2 explains the detailed operation and test methods of the proposed algorithm, Section 3 explains the performance verifications for securing the accuracy and real-time. Finally, concluding remarks with a summary is given in Section 4.

## 2. Materials and Methods

### 2.1. Sample Preparation

A total of 120 sheets of seaweed were used as samples. These include 60 samples of commercial products and 60 samples of seaweed obtained from the real manufacturing site. The size of seaweed is classified into two types according to its shape. The rectangular-shaped seaweed has a length of about 26 cm, and the square-shaped seaweed has a length of about 20 cm. Both types of seaweeds are about 20 cm wide. Among the seaweeds obtained at the real manufacturing sites, 30 samples are containing foreign objects in the manufacturing process. For commercial products, 30 samples of seaweeds are mixed with foreign objects (e.g., insect, shrimp shell, thread, feather, plastic bag, etc.) mainly generated in the actual field to reproduce seaweeds with foreign objects. Finally, 60 samples of normal seaweed and 60 samples of seaweed containing foreign objects were used in the experiment. In Figure 1, two types of seaweed and foreign substances can be seen.

### 2.2. Equipment

The specifications of the push-broom based VNIR hyperspectral camera (FX-10e, Specim, Inc., Oulu, Finland) [20] used in the experiment can be checked in Table 1. 

As a light source, three bulb halogen lamp with a wavelength range of 400 to 1000 nm was used. When the light source was not used, the characteristics of the sample hardly appeared, and when the 1-channel light source was used, it was affected by noise. When a 2-channel light source was used, it was rarely affected by noise, and the characteristics of the sample were well represented. Table 2 shows the light intensity and acquired spectral characteristics of seaweed by the number of channels of the light source.

The hyperspectral imaging system used in the experiment can be seen in Figure 2 The *X*-axis of the hyperspectral camera is fixed at the center and can be adjusted with the *Y*-axis. The light source used 2 channels, the angle is adjustable, and it can be moved along the *Y*-axis. The optimal environment can be established by adjusting the light source and camera. The interior of the dark chamber is light-blocked, allowing the same environment to be maintained at all times. The conveyor belt is the same color used in the industry and is driven at 30 cm/s, which is the same as the speed of the real manufacturing site.

### 2.3. Hyperspectral Characteristics 

Seaweed is a mixture of various materials, so it has a slightly different various color distribution. In addition, seaweed has an uneven and greasy surface. For these reasons, different spectral characteristics can be obtained even when the same amount of light is irradiated. Figure 3 shows the raw spectral characteristics of 10,000 pixels obtained from seaweed and conveyor belt. (The DN of the *Y*-axis is a digital number, which means the intensity of the spectral value.) The spectral characteristics of representative seaweed are shown in Table 2. However, the spectral characteristics of seaweed change due to the color difference and diffuse reflection. When diffuse reflection occurs, the DN value of the wavelength of 500–680 nm increases, and the maximum DN value increases. Therefore, despite the spectral characteristics of pixels obtained from the same seaweed, the offsets between pixels are large. In addition, the spectral characteristics of seaweeds with low DN value distributions have similar spectral characteristics to those of a conveyor belt. To solve this problem, proper calibration is required.

#### 2.3.1. Hyperspectral Image Calibration

In this study, three methods were tested to find the optimal calibration method [21]. Before proceeding with the test, the white-reference value used in the test was obtained through the white balance color checker of x-rite Inc. [22], and the dark current value (D) was obtained by covering the camera cover to block the light. The dark current value was used to remove the noise present in the camera. Table 3 shows the results of the three calibration methods. The spectral characteristics used in the three methods were dark current-removed. (1) Per-Norm [23] is a method of normalizing the spectral characteristics in percentage. Per-Norm applied value ($P_{n}$) can be obtained by removing the minimum value respectively when dividing the spectral value by the maximum value ($r_{max}$). (2) Max-Norm is a method of dividing the spectral values by the maximum value. Finally, (3) Reflection-Norm is a method of dividing the spectral values by the white reference ($W_{ref}$). In general, when processing hyperspectral images, the Reflection-Norm method is popularly used because the effect of the light source is minimized, and the spectral characteristics of the sample can be viewed more accurately. In this case, the Per-Norm method is more effective. The offset between pixels was significantly reduced while maintaining the spectral characteristics of seaweed. Furthermore, since the seaweed and conveyor belt are clearly distinguished, Per-Norm was used as a calibration method. On the other hand, Reflection-Norm cannot be used because even seaweed and conveyor belt pixels are not clearly distinguished.

#### 2.3.2. Dimensionality Reduction

Dimensionality reduction is essential in order to process massive amounts of hyperspectral data in real-time [24,25]. There are several methods of dimensionality reduction, but it is important to maintain the characteristics as much as possible in order to be prepared for unknown foreign objects. In this study, we used the spectral bining function included in the camera. This is a simplified function that acquires only 1/n of the data. For example, if use 1/2 binning, skip the data one by one and acquire it. Even if the camera does not have this function, it can be easily implemented. When applying 1/2 binning, the amount of data to be processed is reduced by 34,406,400 bytes per line, and when applying 1/4 binning, 51,609,600 bytes per line is reduced. In other words, it significantly reduces calculation overhead and enables real-time inspection. Figure 4a–c shows the raw data, shows 1/2 binning, and 1/4 binning results, respectively. The application of 1/4 binning loses some of the spectral characteristics for the peak. Therefore, we used 1/2 binning and confirmed that it achieves real-time data processing.

### 2.4. Proposed Algorithm

The proposed detection algorithm proceeds in 2-stages. Firstly, the conveyor and seaweed are clearly distinguished by subtraction method. Secondly, detailed foreign object inspection is carried out precisely by standardization inspection. All the operations used in the algorithm are simplified and burdenless calculations such as subtraction, division, and one-by-one matching, so it can even run high-speed inspections (30 cm/s) in real-time. 

#### 2.4.1. Distinguishment between Seaweed and Conveyor Belt

In the first stage, the subtraction method uses the difference in values between two different wavelengths in each wavelength range (visible range (400–750 nm) and near-infrared range (750–1000 nm). The subtraction value allows two different subtraction images to be obtained and it also clearly distinguishes seaweed (target) and conveyor belt (background) through threshold setting. Even if the target object changes, you can change the band combination and threshold settings to easily distinguish the target from the background. The average of the values obtained from Table 3 (Per-Norm applied data) was used as reference data for seaweed and conveyor belt. Firstly, the wavelength value with the largest difference of subtraction value between seaweed and conveyor belt shall be selected. We calculated all cases in an empirical way to find the best band combination with the largest difference between the subtraction value of seaweed and conveyor belt. As a result, the 26th (532.25 nm) and 63rd (731.79 nm) bands ($P_{26}$ and $P_{63}$) are selected in the visible range ($S_{V})$, and the 68th (759.14 nm) and 93th (897.28 nm) bands ($P_{68}$ and $P_{93}$) are selected in the near-infrared range($S_{N}$), respectively. When seaweed and conveyor belts can be clearly distinguished, there are two benefits to this operation. Firstly, it significantly reduces unnecessary calculations by ignoring the standardization inspection of the conveyor belt part (Figure 5 shows ignored zone). This is because foreign objects on the conveyor belt do not need to be detected. The ignored zone is about 20% of the total seaweed area, which can save a lot of computation. Secondly, it is possible to apply operations optimized for each of the seaweed pixels and conveyor belt pixels. In addition, accuracy also be improved by using 2-class classification instead of 3-class classification. Two types of subtraction images can be seen in Figure 6b,c.
(1)$$S_{V}={\;P}_{63}-P_{26}\;S_{N}={\;P}_{93}-P_{68}$$
where $S_{V}$ is subtracted value of the visible range, and $S_{N}$ is subtracted value of the near-infrared range.

#### 2.4.2. Detection of Foreign Object

In the second stage, the standardization inspection is used for the detection of foreign objects and inspects after standardizing the pixels. The standard deviation for each seaweed and conveyor belt was calculated from the data in Table 3 (Per-Norm applied). The equation of standardization is shown in Equation (2). After standardizing pixels, the offset between the pixels is dramatically reduced.
(2)$$x_{new}=\frac{x-\mu }{\sigma }$$
where μ is mean value of x, and σ is standard deviation of x.

### 2.5. Multi-Class Support Vector Machine

Multi-class Support Vector Machine (MCSVM) was adopted to compare the performance with the proposed algorithm. SVM [26] is a supervised learning model as one of the fields of machine learning, and is mainly used for classification and regression analysis. Basically, SVM is a binary classifier and a linear classifier, but it can be extended as a multi-class classifier and a nonlinear classifier. In order to perform nonlinear and multi-class classification, data must be mapped into a high-dimensional feature space. Kernel tricks are used to do this efficiently. In this experiment, we used the Spectral Angle Mapper (SAM) kernel-based SVM to test the algorithm. SAM calculates the angle in the n-dimensional space between the trained and test pixels. Classify through this angle. SAM kernel is widely used in hyperspectral imaging because it can quickly measure and classify the similarity of spectral angles [27,28,29,30]. The equation of SAM is shown in (3). We used the same input as the proposed algorithm as the input of SAM (Per-Norm and 1/2 binning applied).
(3)$$\theta \left(x,y\right)=cos^{-1}\left(\frac{\sum_{i=1}^{n}x_{i}y_{i}}{{\left(\sum_{i=1}^{n}x_{i}^{2}\right)}^{\frac{1}{2}}\times {\left(\sum_{i=1}^{n}y_{i}^{2}\right)}^{\frac{1}{2}}}\right)$$
where x is spectral value of test pixel, *y* is spectral value of trained pixel, and *n* is the number of bands.

## 3. Results and Discussion

### 3.1. Detection Results of Proposed Algorithm

In the experiment, pre-obtained hyperspectral images were used to test the performance of the proposed algorithm. This is the reason for measuring quantitative values in the same environment. Before applying the algorithm, the hyperspectral image was calibrated through Per-Norm, and the 1/2 binning function was used to reduce the number of bands in half (112 bands). 

#### 3.1.1. Subtraction Method

The subtraction image in Figure 6b,c shows that seaweed and conveyor belts are clearly classified. Threshold settings in the visible light range, the seaweed has a subtraction value of 60 to 80 and a conveyor belt has a subtraction value of 20 to 45, respectively. And in the near-infrared range, seaweed has a subtraction value of −30 to −5 and a conveyor belt has a subtraction value of 5 to 25, respectively. According to this result, thresholds were used to distinguish seaweed and conveyor belts. In addition, the remaining values are classified as foreign objects. It can be seen that the shrimp shell, which is relatively easy to distinguish, can be detected through the subtracted image in the visible light range. Seaweed and conveyor belts that are well classified in both the visible and near-infrared ranges, follow the results of the visible light range. In the case of foreign objects, it is classified as a foreign object if it is found in either of both ranges. Figure 7 shows the classification results through the subtraction method.

#### 3.1.2. Standardization Inspection

The results of applying standardization to reference data (Table 3 Per-Norm applied data) are shown in Figure 8. The limit range can be specified by giving margins to the maximum and minimum values of each band. In this experiment, a margin of 3 was given. If a value is outside the limit range, the pixel is considered a foreign object. Reducing the margins can increase sensitivity while increasing margins can reduce sensitivity. As an example, to test the performance of the standardization inspection [31], a piece of black plastic bag was used as a foreign object. Figure 9a raw hyperspectral image shows that it is difficult to detect a piece of black plastic bag even with the naked eyes. Figure 9b shows that plastic bag detection failed by subtraction method. Some pixels of black plastic bags were classified as seaweed and some as conveyor belts. After applying standardization, the pixel values of the black plastic bag are shown in Figure 10. In seaweed pixels, values out of range occurred near 530–700 nm and 880–100 nm wavelengths, and in conveyor belts pixels, values out of range occurred near 400 nm and 780 nm wavelengths. Finally, the successful detection of a piece of black plastic bag is shown in Figure 8, and the overall process of the proposed algorithm is shown in Figure 11.

### 3.2. Detection Results of MCSVM

The multi-class support vector machine (MCSVM) used in the experiment is a three-class (Seaweed, conveyor belt, and foreign objects) classifier using a 112-dimensional SAM kernel. As an example, to test the performance of MCSVM, we used the same hyperspectral image used in the proposed algorithm. Both shrimp shells and black plastic bags were successfully detected. However, it can be seen from Figure 12a that many FP (False-Positive) pixels have occurred.

### 3.3. Detection Performance of Each Algorithm

To measure the quantitative performance of the proposed algorithm, 60 normal seaweed samples and 60 seaweed samples containing foreign objects were used, and 2 kinds of seaweeds were used. If there are pixels classified as foreign objects at least 1 pixel in the position of foreign objects, the detection is considered successful (True-Negative (TN)). In contrast, if at least 1 foreign object pixel is found in normal seaweed, it is considered misclassification (FN). The MCSVM has excellent foreign object classification performance. However, a number of FN have occurred. If about 30% of normal seaweed is rejected, it will have a fatal effect on productivity. In addition, the processing speed per line is about 60 ms, so real-time inspection cannot be performed. The proposed algorithm also shows prominent detection performance. FN has not occurred, and the processing speed per line is about 3 ms, which can operate in real-time even at the maximum frame rate (330 fps). Considering the various performances, the proposed algorithm is suitable for applications in real manufacturing sites. The detection performance of each algorithm can be seen in Table 4.

### 3.4. Foreign Object Detection Platform

The foreign object detection platform was implemented using the proposed algorithm to verify its applicability in the real manufacturing site. It was implemented in the Visual studio 2017 MFC environment and used the C++ language. After classifying the pixels using the proposed algorithm, the display color is converted (seaweed—black, conveyor belt—white, foreign object—red) to increase the visibility of the classified results. Through this platform, we confirmed that the proposed algorithm works well at maximum fps. The platform can be seen in Figure 13.

## 4. Conclusions

The purpose of this study is to confirm the possibility of rapid and accurate foreign object detection using hyperspectral imaging technique. The proposed algorithm uses a simplified operations and dimensionality reduction for real-time inspection. In addition, efficiency and accuracy have been improved through a 2-stage detection method. In the first stage, the subtraction method distinguishes seaweed (target image) and conveyor belts (background image) very quickly and accurately by extracting the features of each of the visible and near-infrared ranges. In the visible range, the wavelength of 532.25 nm (26th band) and 731.79 nm (63rd band) were selected, and in the near-infrared range, the wavelength of 759.14 nm (68th band) and 897.28 nm (93rd band) were selected. At this stage, foreign objects that are relatively easy to detect can also be detected. Through this, unnecessary operations can be eliminated by excluding the conveyor belt part and foreign object pixels from the next inspection stage. In the second stage, the standardization inspection more carefully detects foreign objects which are difficult to detect by comparing the reference data by one-by-one matching. As a result of the experiment, the detection accuracy of proposed algorithm was 95%, and the recall rate reached 100%. The MCSVM used to compare the performance showed excellent foreign object detection performance, but a number of FN occurred. Additionally, MCSVM cannot perform rapid inspection in real-time, but the proposed algorithm can perform a real-time inspection even at maximum fps, which has been revealed through platform implementation. Since the proposed algorithm easily compares and classifies the hyperspectral images of arbitrary foreign object and food, we expect that our work will be helpful in securing the safety of various food groups.

For future work, we plan to confirm the possibility of faster inspection through improved camera specifications. Currently, faster inspections are not possible due to camera specifications, but if the camera specifications are improved (higher maximum FPS), faster inspection is possible using higher dimensionality reduction (1/4 binning). Rapid inspection technique using hyperspectral image can be applied not only in food products but also in other industries. Finally, we plan to explore the applicability of the proposed algorithm to various manufacturing industries.

## Figures and Tables

**Figure 1 sensors-21-05279-f001:**
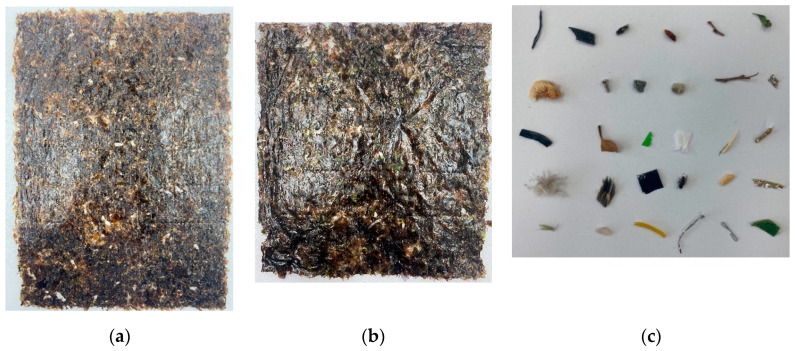
Seaweed used in the experiment. (**a**) Obtained from the real manufacturing site (Rectangular-shaped), (**b**) Commercial products (square-shaped) (**c**) Examples of foreign objects.

**Figure 2 sensors-21-05279-f002:**
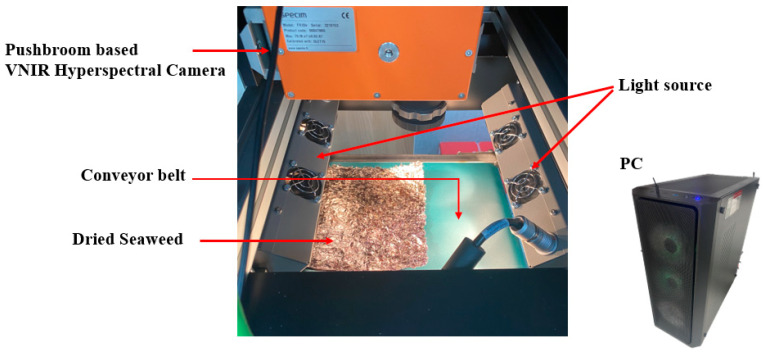
The VNIR hyperspectral imaging system used in the experiment.

**Figure 3 sensors-21-05279-f003:**
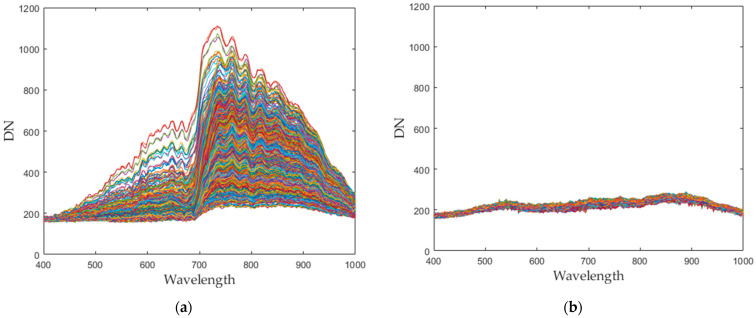
Spectral characteristics of 10,000 pixels (**a**) seaweed, (**b**) conveyor belt.

**Figure 4 sensors-21-05279-f004:**
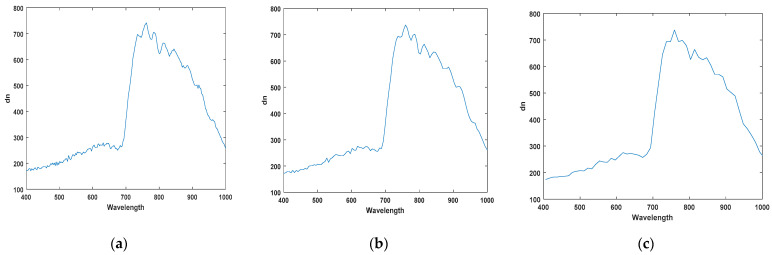
Spectral characteristics for each step of spectral binning (**a**) Full bands (224 bands) (**b**) 1/2 binning applied (112 bands), (**c**) 1/4 binning applied (56 bands).

**Figure 5 sensors-21-05279-f005:**
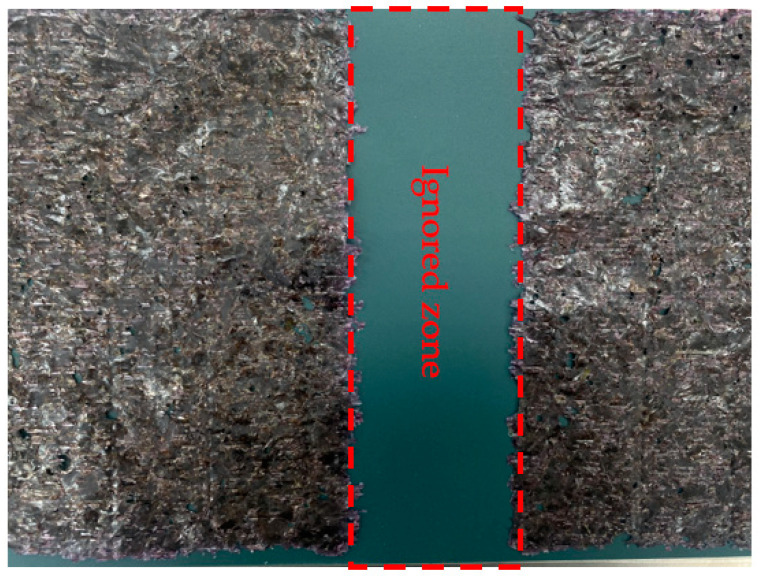
Example of an ignored zone.

**Figure 6 sensors-21-05279-f006:**
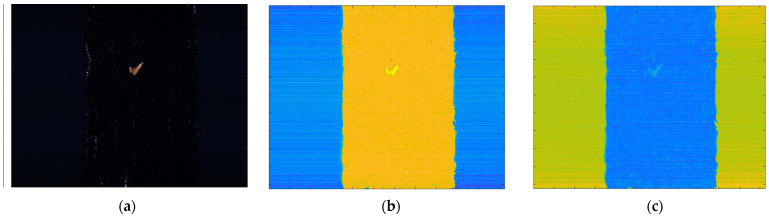
(**a**) Raw image, (**b**) Subtraction image in the visible range (**c**) Subtraction image in the near-infrared range.

**Figure 7 sensors-21-05279-f007:**
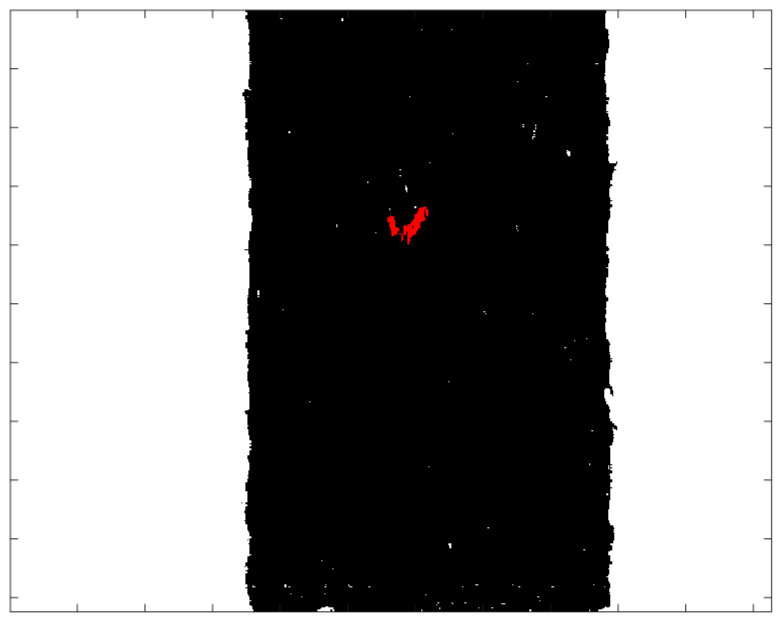
Classification result of the subtraction method.

**Figure 8 sensors-21-05279-f008:**
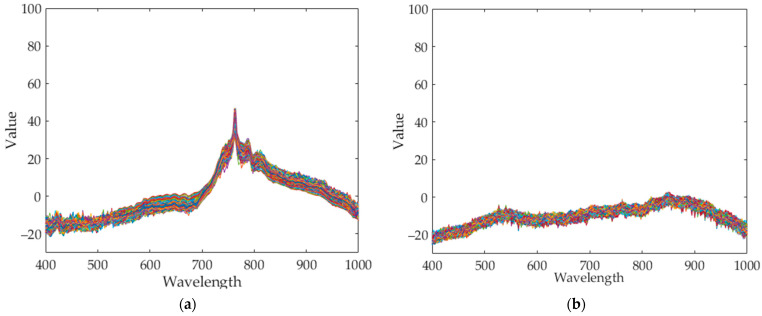
Result of standardization (**a**) Seaweed (**b**) Conveyor belt.

**Figure 9 sensors-21-05279-f009:**
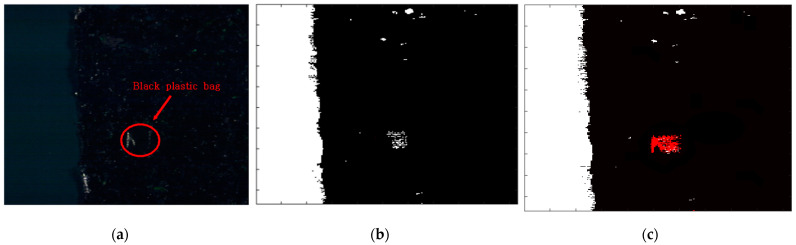
Results of each inspection step. (**a**) Raw hyperspectral image, (**b**) After applying the subtraction method (**c**) After applying the standardization inspection.

**Figure 10 sensors-21-05279-f010:**
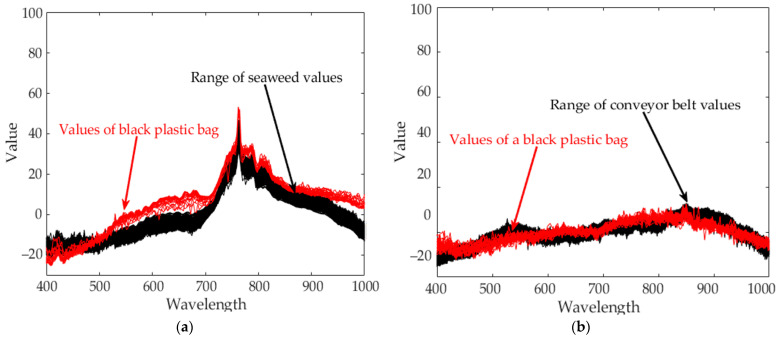
The range of the reference data and the value of the black plastic bag obtained after standardizing. (**a**) About seaweed pixels (**b**) About conveyor belt pixels.

**Figure 11 sensors-21-05279-f011:**
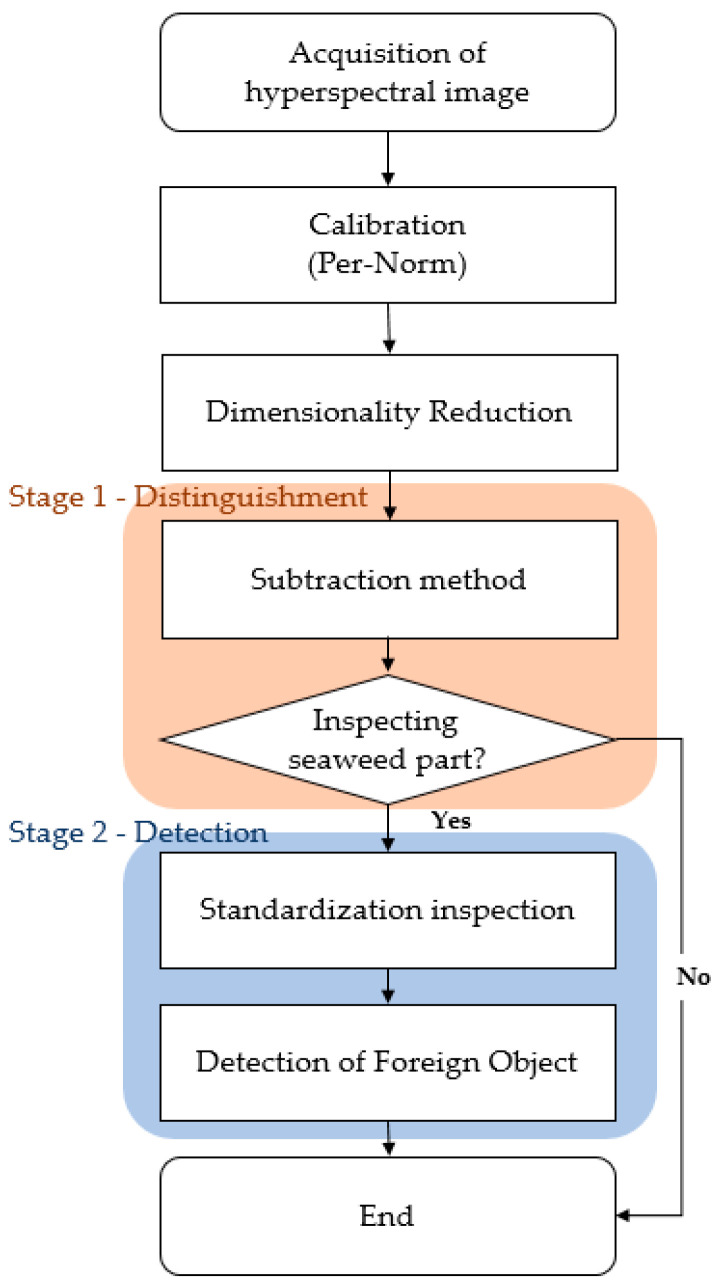
Flow chart of overall process.

**Figure 12 sensors-21-05279-f012:**
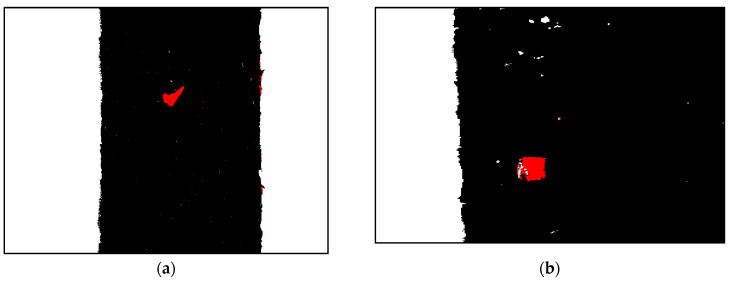
Classification result of the MCSVM. (**a**) Shrimp Shell (**b**) Black plastic bag.

**Figure 13 sensors-21-05279-f013:**
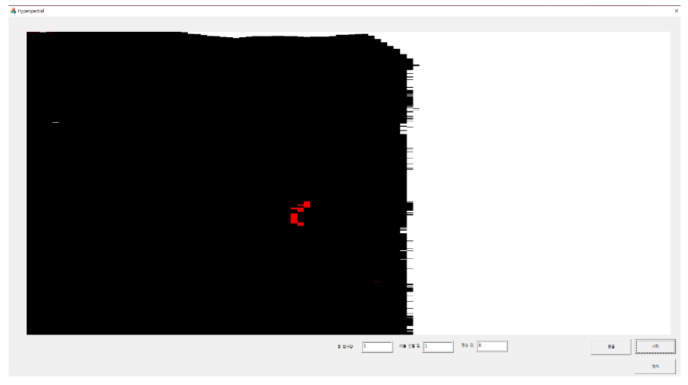
Foreign object detection platform.

**Table 1 sensors-21-05279-t001:** Specifications of the FX-10e.

Parameters	Values
Spectral range	400–1000 nm
Spectral bands	224 bands (Maximum)
Spatial resolution	1024 pixel
Spectral resolution	2.7 nm
Exposure time	3.22 ms
Frame rate (Line rate)	330 fps (Maximum)

**Table 2 sensors-21-05279-t002:** Light intensity and spectral characteristics by number of light source channels.

Channels	0 ch	1 ch	2 ch
Intensity	10 lux	1100 lux	2050 lux
Spectral Characteristics	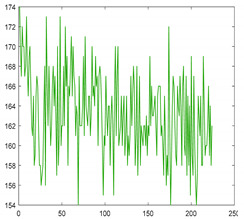	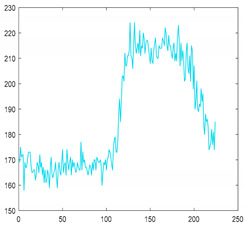	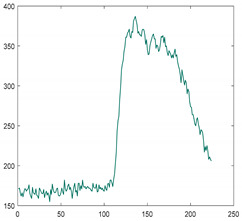

**Table 3 sensors-21-05279-t003:** Calibration results.

Method	Per-Norm	Max-Norm	Reflectance-Norm
Seaweed	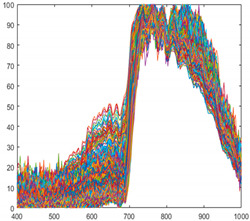	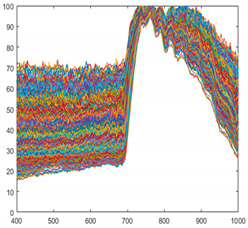	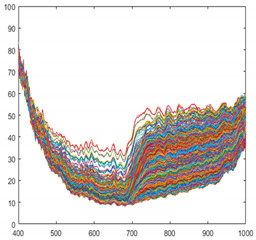
Conveyor belt	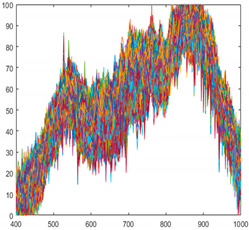	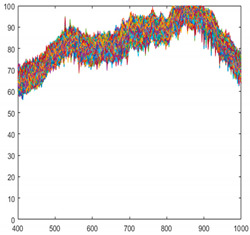	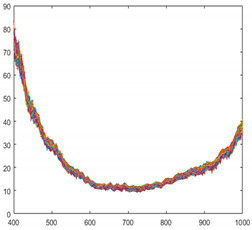
Equation	$P_{n}=\frac{\left(r_{n}-r_{min}\right)-D}{(r_{max}-r_{min})-D}\times 100$ where $r_{n}$ is a nth spectral value	$M_{n}=\frac{r_{n}-D}{r_{max}-D}\times 100$	$R_{n}=\frac{\left(r_{n}-D\right)}{\left(W_{ref}-D\right)}$

**Table 4 sensors-21-05279-t004:** Detection performance of each algorithm.

Method	Detection Result				Performance			Processing Time/Line
	TP	FN	FP	TN	Recall	Precision	Accuracy	
Proposed algorithm	60	0	6	54	1	0.91	0.95	3 ms
MCSVM	39	21	4	56	0.65	0.91	0.79	60 ms

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
