# Peer review of "Rapid Foreign Object Detection System on Seaweed Using VNIR Hyperspectral Imaging"

_sensors, 2021, doi:10.3390/s21165279_

Round 1

Reviewer 1 Report

This work proposed a novel algorithm to detect foreign objects in hyperspectral food images. The proposed algorithm achieved promising results, and the experiment performed suggests that it could be employed in real-world settings. This is a very interesting work, but there are some issues that should be improved. Such issues are listed below:

  • The authors mentioned that previous works (references 6-10) proposed the use of hyperspectral images to detect foreign objects in food. Please consider including a brief discussion on these works, especially concerning their execution time which is a major limiting factor for real-time use. 
  • Presenting an equation as part of Table 3 may be confusing. Please consider presenting this equation in the text.
  • The first step of the proposed algorithm is not very clear. Please consider re-writing the first paragraph of section "2.4.1. Distinguishment between Seaweed and Conveyor belt". It seems that equation 1 summarizes this whole process, i.e., SV and SN are the results of this step. Is this correct? If so, please consider summarizing this at the beginning of the section (maybe extending the first sentence), i.e., state that two different images are obtained in this first step: one referent of the visible range and another referent to the infrared range. Then the following statements would explain how SV and SN are obtained.
    Moreover, the authors may consider including references to Figures 6b and 6c in section "2.4.1. Distinguishment between Seaweed and Conveyor belt", in order to provide an easier understanding of these operations. 
  • The second step of the proposed algorithm (in section "2.4.2 Detection of Foreign Object") seems to consist of signal processing followed by a thresholding process. Is this correct? The authors mentioned that a range is the criteria used to define the pixels that are related to foreign objects. If possible, specify this range. Which are the limit values that define if a pixel can be considered as part of a foreign object? In section "3.1.2 Standardization Inspection", the authors mentioned some values, but it is not clear if these are the limits/thresholds. In case such values are the thresholds used in step 2 (described in section "2.4.2 Detection of Foreign Object"), please include these values in section  2.4.2.
    It seems that the result of this step is exemplified in Figure 7. Is this correct? If it is so, please include a reference to Figure 7 in section 2.4.2. This would provide a quick visual example of the result of this step.
  • In section "2.5 Multi-Class Support Vector Machine", please specify the input of the SVM classifier. The authors stated the features used as input were calculated using SAM. Was SAM applied to the original hyperspectral image, or was any pre-processing carried out?  Please, explain this classification process in more detail.
  • Finally, evaluate if part of the information presented in the "3. Results and Discussion" section should not be included as part of the "2. Materials and Methods" section. Some information seems to be out of place.

Reviewer 2 Report

The authors propose an algorithm for detecting foreign objects in seaweed used for food.

Hyperspectral technology will obviously give more accurate results than visible technology.

In addition, the authors seek to increase productivity by subtracting the reflection from the conveyor belt from the processed signal. This method of subtracting a signal that changes little over a portion of the frame is often used in image compression, for example, in jpeg technology. It really gives good results.

The response of different varieties of algae is presented by the authors as a vector in a multidimensional space. This is clear.

It is not clear why the authors chose the wavelengths of 532.25 nm (26th range) and 731.79 nm (63rd range) to extract characteristics in the visible and near infrared ranges, and wavelengths 759 were chosen in the near infrared range, 14 nm (68th range) and 897.28 nm (93rd range). This corresponds to some characteristics of the research object. How?

It seems convincing that the algorithm proposed by the authors, presented in figure 10, gives obvious positive results.

Although the reviewer does not have the opportunity to verify the developers' data by direct experiment or calculations, the results presented by the authors seem to be quite reliable.

The only thing that, in my opinion, could be added as part of the continuation of this work, is to look at the statistics of the appearance of various foreign objects in order to tune the training system to a greater extent for more frequently appearing objects. And even better - on the most dangerous to health.

In general, the authors show the possibility of increasing the productivity up to that required in the production plants for food processing of algae. However, the 5% chance of missing a foreign object is of course small for fully automated production. I would not like to buy these products and receive an inedible item in each of the 20 cans. I would like to wish the authors to bring their work to the full automation of the process.
